Subject Areas:
physiology/biochemistry/evolution

Keywords:
Hemiptera, Heteroptera, juvenile hormone, juvenile hormone III skipped bisepoxide, true bugs, ultra-performance liquid chromatography coupled with tandem mass spectrometry

Author for correspondence:
Shin G. Goto
e-mail: shingoto@osaka-cu.ac.jp

# Juvenile hormone III skipped bisepoxide is widespread in true bugs (Hemiptera: Heteroptera)

Keiji Matsumoto[1], Toyomi Kotaki[2], Hideharu Numata[3], Tetsuro Shinada[4] and Shin G. Goto[1]

[1]Department of Biology and Geosciences, Graduate School of Science, Osaka City University, 3-3-138 Sugimoto, Sumiyoshi-ku, Osaka 558-8585, Japan
[2]Institute of Agrobiological Sciences, National Agriculture and Food Research Organization, Tsukuba, Ibaraki, Japan
[3]Department of Zoology, Graduate School of Science, Kyoto University, Kyoto, Japan
[4]Department of Material Science, Graduate School of Science, Osaka City University, Osaka, Japan

KM, 0000-0001-5580-3802; TK, 0000-0002-1830-5457;
HN, 0000-0003-3786-0701; TS, 0000-0001-9145-1533;
SGG, 0000-0002-4431-7531

Juvenile hormone (JH) plays important roles in almost every aspect of insect development and reproduction. JHs are a group of acyclic sesquiterpenoids, and their farnesol backbone has been chemically modified to generate a homologous series of hormones in some insect lineages. JH III (methyl farnesoate, 10,11-epoxide) is the most common JH in insects, but Lepidoptera (butterflies and moths) and 'higher' Diptera (suborder: Brachycera; flies) have developed their own unique JHs. Although JH was first proposed in the hemipteran suborder Heteroptera (true bugs), the chemical identity of the heteropteran JH was only recently determined. Furthermore, recent studies revealed the presence of a novel JH, JH III skipped bisepoxide ($JHSB_3$), in some heteropterans, but its taxonomic distribution remains largely unknown. In the present study, we investigated $JHSB_3$ production in 31 heteropteran species, covering almost all heteropteran lineages, through ultra-performance liquid chromatography coupled with tandem mass spectrometry. We found that all of the focal species produced $JHSB_3$, indicating that $JHSB_3$ is widespread in heteropteran bugs and the evolutionary occurrence of $JHSB_3$ ascends to the common ancestor of Heteroptera.

# 1. Introduction

Hemiptera is the largest order of hemimetabolous insects comprising four suborders: Sternorrhyncha, Auchenorrhyncha (including the infraorders Cicadomorpha and Fulgoromorpha), Coleorrhyncha and Heteroptera [1–3]. Heteroptera includes seven infraorders, i.e. Enicocephalomorpha, Dipsocoromorpha, Leptopodomorpha, Gerromorpha, Nepomorpha, Cimicomorpha and Pentatomomorpha [1,2]. Heteropterans have evolved diverse life histories and specialized morphological adaptations enabling them to colonize terrestrial, semiaquatic and aquatic habitats. Furthermore, heteropterans exploit various food sources including plants, fungi and animals, including vertebrate blood [2,4]. Among the suborders of Hemiptera, Heteroptera is the most diversified, representing more than 45 000 described species [1]. Based on an estimated 1.55 million described species of animals, Heteroptera represents 2.9% of animal diversity [1,3].

Wigglesworth [5,6] discovered a humoral factor maintaining juvenile characters in the blood-sucking heteropteran *Rhodnius prolixus* (family: Reduviidae; assassin bugs). Later he referred to this factor as the juvenile hormone (JH) [7]. JH plays important roles in almost every aspect of insect development and reproduction, including metamorphosis, caste determination in social insects, regulation of behaviour, polyphenism, larval and adult diapause regulation, ovarian development and various aspects of metabolism associated with these functions [8,9]. JHs are a group of acyclic sesquiterpenoids, and their farnesol backbone has been chemically modified to generate a homologous series of hormones, such as JH 0, JH I, JH II and JH III with a single epoxide and JHB$_3$ with two epoxides [8]. JH III is the most widely distributed JH in insects, whereas JH 0, I and II are found only in Lepidoptera (butterflies and moths), and JHB$_3$ is found only in 'higher' Diptera (suborder: Brachycera; flies) [8] (figure 1).

Although the presence of JH was first reported in Heteroptera, the chemical structure of heteropteran JH remained controversial for a long time [10–14]. The chemical structure of heteropteran JH was finally clarified in a stink bug (family: Pentatomidae) as methyl (2R,3S,10R)-2,3;10,11-bisepoxyfarnesoate, which was named JH III skipped bisepoxide (JHSB$_3$) [15,16]. Recent studies also revealed the occurrence of JHSB$_3$ in stink bugs [17], a bean bug (family: Alydidae) [18] and assassin bugs [19]. Its biological activity was also revealed in stink bugs and a bean bug [16–18]. By contrast, the JH present in some sternorrhynchan aphids (family: Aphididae) and a whitefly *Bemisia tabaci* (family: Aleyrodidae), and a fulgoromorphan planthopper *Nilaparvata lugens* (family: Delphacidae) were determined to be JH III [20–24]. These studies indicated that the ancestor of Hemiptera used JH III as their innate JH, but a certain heteropteran lineage developed JHSB$_3$ and began using it as a JH. This was a defining event in insect evolution, but which heteropteran lineage switched the JH system remains unclear owing to the limited and fragmentary nature of our current understanding of heteropteran JHs.

To approach this issue, we investigated the corpus allatum (CA) products of 31 species belonging to 30 genera, 21 families, and five infraorders in Heteroptera, to determine whether these taxa produce JHSB$_3$. The CA is the JH-producing endocrine organ and culturing it allowed us to efficiently collect JH with low contamination. A highly sensitive analytical method was also required because some species are small, and their CAs were expected to produce a small amount of JH. We used ultra-performance liquid chromatography coupled with tandem mass spectrometry (UPLC-MS/MS) with a chiral column. JHSB$_3$ [(2R,3S,10R)-form] and its stereoisomers [(2R,3S,10S), (2S,3R,10R) and (2S,3R,10S)-forms] are characterized by the presence of distal epoxides. As the ordinal C18 column does not discriminate between JHSB$_3$ and its stereoisomers, analysis with a chiral column is necessary for the identification of JHSB$_3$ [15,18]. Our chiral UPLC-MS/MS could detect JHSB$_3$ at the pg level [19]. In the present study, we also focused on the presence of 10S-JHSB$_3$, which was recently identified as a heteropteran JH [17].

# 2. Material and methods

Insects were collected from the field in Osaka, Nara, Kyoto and Okayama prefectures in Japan or obtained from colonies in our and other laboratories (see electronic supplementary material, table S1 and figure S1). Only adults were used. Due to the limited number of individuals collected from the field, results from 14 species are based on a single specimen. Insects were individually anaesthetized on ice and immobilized by clay. The CA attached with the corpora cardiaca was removed from these individuals according to the methods outlined in a previous study [25]. In brief, the CA was incubated in 30–50 µl of the modified minimal essential medium at 30°C for 5 h. The JHs were extracted with hexane, dried under the stream of argon gas and dissolved again in 30–1200 µl of methanol (electronic supplementary material, table S1).

**Figure 1.** Structures of various JHs.

The UPLC-MS/MS (ACQUITY UPLC H-Class, Xevo TQ-S micro, Waters, Milford, MA, USA) with a chiral column (CHIRALPAK IA-U, $3.0 \times 100$ mm, 1.6 µm particle size, Daicel, Tokyo, Japan) was used to compare the retention times [18,19]. Authentic JHSB$_3$ and 10S-JHSB$_3$ were synthesized as described previously [15]. The MS/MS analysis of the authentic JHSB$_3$ showed the $[M + H]^+$ ion at $m/z$ 283.2 and the $[M + Na]^+$ at $m/z$ 305.3. The product ions were detected at $m/z$ 42.9 and $m/z$ 233.2 when ions at $m/z$ 283.2 were used as a precursor, whereas no fragmentation was detected when ions at $m/z$ 305.3 were used. In the present study, ions at $m/z$ 283.2 and their product ions at $m/z$ 233.2 were used as monitor ions for detecting JHSB$_3$ and 10S-JHSB$_3$. The lowest detection limit of JHSB$_3$ in our methodology is 0.25 pg [19]. Ten microlitres of the CA product in methanol was used.

The UPLC data obtained are available in the Dryad Digital Repository, https://doi.org/10.5061/dryad.z08kprr9f.

## 3. Results

We did not examine the species belonging to Dipsocoromorpha and Enicocephalomorpha, simply because we were unable to collect them in the field. However, we successfully collected species from other heteropteran lineages (figure 2 and electronic supplementary material, table S1): three species from Gerromorpha, two from Nepomorpha, one from Leptopodomorpha, eight from Cimicomorpha and 17 from Pentatomomorpha. We analysed their CA products with the chiral UPLC-MS/MS. It is important to note that the retention time of JHSB$_3$ was often shifted under the analytical conditions. Therefore, we clarified the retention time of the authentic JHSB$_3$ immediately preceding the analysis of natural JH in every analytical session and present the data obtained from each session separately.

Figure 3 shows the results of the UPLC-MS/MS analyses of the CA product. In all species tested, the retention time of the main peak of the CA product was identical to that of the authentic JHSB$_3$, showing that all tested species produced JHSB$_3$. In addition to the major JHSB$_3$ peak, a peak corresponding to the 10S-JHSB$_3$ was observed in some species, such as *Plautia stali*, *Riptortus pedestris*, *Dolycoris baccarum*, *Geocoris jucundus* and *Cletus punctiger*, but the peak was small, being almost at the detection limit, and we could not conclude whether these species synthesized 10S-JHSB$_3$.

R. Soc. Open Sci. **8**: 202242

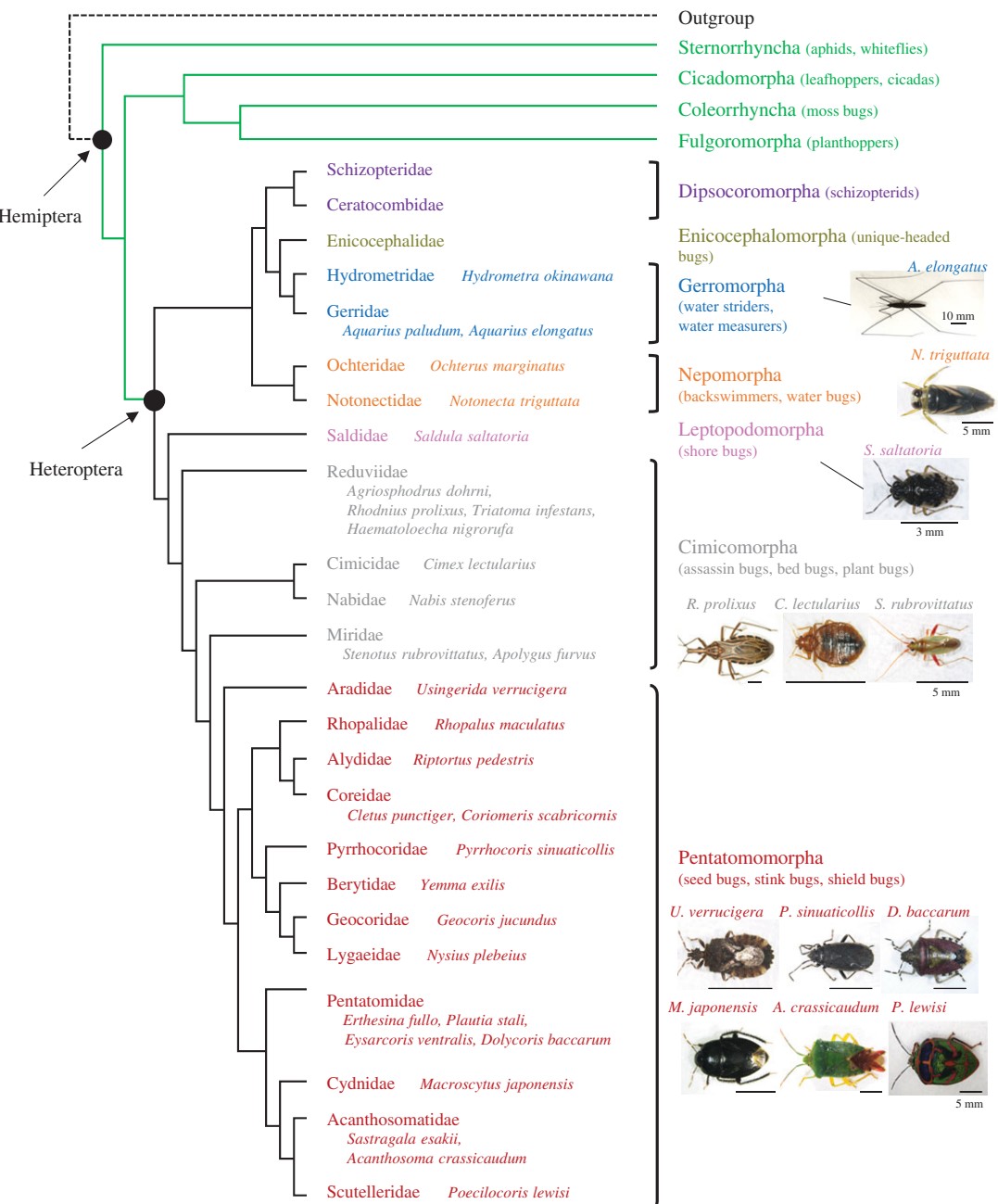

**Figure 2.** Species tested in the present study mapped on the hemipteran phylogenetic tree inferred from a previous mitogenomic analysis [26]. An outgroup included Psocidae, Lepidopsocidae, Ectobiidae, Rhinotermitidae and Mantidae. Photographs of some species tested in the present study are also shown. The common ancestors of Hemiptera and Heteroptera are marked by closed circles.

## 4. Discussion

The common ancestor of Hemiptera diversified into Sternorrhyncha and the remaining Hemiptera at the end of the radiation of spermatophytes (seed plants) 309 Ma in the Late Carboniferous period [26,27]. The Permian diversification of hemipterans immediately followed with the origin of Heteroptera 290–268 Ma, which then diversified in the Late Permian and Triassic [26,28]. The Permian–Triassic mass extinction event, during which many niches became available for resource partitioning, possibly contributed to the acceleration of heteropteran diversification [26,29]. In the present study, we clarified that all species belonging to Pentatomomorpha, Cimicomorpha, Leptopodomorpha, Nepomorpha and Gerromorpha produces $JHSB_3$ (figures 2 and 3), suggesting that the common ancestor of Heteroptera possessed $JHSB_3$ as an innate JH.

**Figure 3.** Chiral ultra-performance liquid chromatography coupled with tandem mass spectrometry of the corpus allatum products from various heteropteran species. The vertical axis indicates the signal intensity of the product ion at *m/z* 233.2 produced from the precursor ion at *m/z* 283.2. Panels (*a*–*l*) are the results of each analytical session. The retention time of authentic JHSB$_3$ was clarified before each session. (*g*) and (*k*) contain a noise peak in the JHSB$_3$ standard (#). Small peaks corresponding to 10*S*-JHSB$_3$ were detected in *Plautia stali*, *Riptortus pedestris*, *Dolycoris baccarum*, *Geocoris jucundus* and *Cletus punctiger* indicated by asterisks (*). Full species names can be found in figure 2 and electronic supplementary material table S1.

The heteropterans that use JHSB$_3$ emerged from hemipteran species that had used JH III [20–24], indicating the evolutionary occurrence of JHSB$_3$ ascends to the common ancestor of Heteroptera [8,26]. Both JH III and JHSB$_3$ possess 10,11-epoxide, but JHSB$_3$ also possesses 2,3-epoxide. Thus far, enzymes

responsible for epoxidation at the two sites have not been elucidated in Heteroptera. In Blattodea (cockroaches) and Lepidoptera, the cytochrome P450 enzymes CYP15A1 and CYP15C1, respectively, convert the 10,11-double bond of the JH III precursor to 10,11-epoxide [30,31]. In the case of JHB$_3$, which possesses two epoxides at the 6,7- and 10,11-positions, a distinct P450 enzyme, CYP6G2, was proposed for the bisepoxidation [32–34]. The common ancestor of Heteroptera might have developed a new P450 that enables epoxidation at both positions or they might have modified the activity of the pre-existing CYP15 to produce two epoxides. A CYP15 that is expressed in the CA was reported in some bugs [35,36]. Investigating the enzymatic activity of the CYP15 and other P450 enzymes is important to clarify the process for JHSB$_3$ synthesis. This is a key evolutionary event that shaped heteropteran physiology.

In the present study, we primarily focused on the presence/absence of JHSB$_3$, but not JH III, in the heteropteran lineages. It is necessary to investigate the presence/absence of JH III in addition to JHSB$_3$ in these heteropteran species in a future study, although recent studies revealed no evidence for the presence of JH III in alydid, lygaeid and pentatomid bugs (Infraorder Pentatomomorpha) and reduviid bugs (Infraorder Cimicomorpha) [12,13,19,36–39]. It is also important to investigate the presence of JHSB$_3$ with much wider taxon sampling in a future study, to find out how far back the origin of JHSB$_3$ can be.

Heteroptera contains many species closely associated with humans. Triatomine bugs (Cimicomorpha; Reduviidae) are the principal vector of the parasite that causes Chagas disease, and bed bugs (Cimicomorpha; Cimicidae) result in skin rashes and allergic symptoms. Plant bugs (Cimicomorpha; Miridae), leaf-footed bugs (Pentatomomorpha; Coreidae), stink bugs (Pentatomomorpha; Pentatomidae) and shield bugs (Pentatomomorpha; Scutelleridae) include many serious agricultural pests [1]. Thus, their management is crucial for economic stability [4]. Synthetic compounds that mimic the action of JHs (JH analogues, JHAs) are in a class of insecticides called insect growth regulators and have been used for several decades [40]. Although the utility of JHAs for insect pest control is limited, the relatively fewer effects of JHAs on non-target insects and other animals and favourable environmental fate of these compounds make them attractive insecticides for inclusion in integrated pest management programmes [40]. All heteropterans use JHSB$_3$, suggesting that JHSB$_3$ will be the primary molecule for developing new insect growth regulators that widely control heteropteran pests while being specific enough to limit non-target effects [41].

In conclusion, we clarified the occurrence of JHSB$_3$ in heteropteran lineages during insect evolution. Although there are estimated to be more than 1 million insect species categorized into 39 orders [3], JHs have been identified in only around 100 species from around 10 orders [8]. Further studies on the orders not previously studied may reveal additional novel JHs.

Data accessibility. Data available from the Dryad Digital Repository: https://doi.org/10.5061/dryad.z08kprr9f [42].

Authors' contributions. K.M., H.N., T.K., T.S. and S.G.G. participated in the design of the study; K.M., H.N. and T.K. collected insects; K.M. carried out the experiments; K.M., T.S. and S.G.G. participated in data analysis; S.G.G. drafted the manuscript; all authors critically revised the manuscript; all authors gave final approval for publication and agree to be held accountable for the work performed therein.

Competing interests. The authors have no potential conflicts of interest to the research, authorship and/or publication of this article.

Funding. This work was supported by the Grant-in-Aid for Scientific Research by the Wellness Open Living Laboratory (WOLL) LLC (for S.G.G. and T.S.).

Acknowledgements. We thank Nobuki Muramatsu (Kyoto University), Hirotaka Kanuka and Erisha Saiki (The Jikei University School of Medicine), Minoru Moriyama (National Institute of Advanced Industrial Science and Technology), Shin-ya Ohba (Nagasaki University), Hiromi Mukai (Forestry and Forest Products Research Institute), and Ayumu Mukai, Yu Suzaki, Genyu Mano, Takaaki Tamai and Naotaka Aburatani (Osaka City University), for providing us insects. We thank Takako Shizuka (Osaka City University) for preparing figures. We also acknowledge Editage (www.editage.co.jp) for English correction.

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
