## [Peer Review File · Royal Society Open Science]

Review History

RSOS-202242.R0 (Original submission)

Review form: Reviewer 1 (Kenneth Davey)

Is the manuscript scientifically sound in its present form?

Yes

Are the interpretations and conclusions justified by the results?

Yes

Is the language acceptable?

Yes

Do you have any ethical concerns with this paper?

No

Have you any concerns about statistical analyses in this paper?

No

Recommendation?

Accept with minor revision (please list in comments)

Comments to the Author(s)

The principal objective of this study is to demonstrate that the Heteroptera are distinguished from the remaining Hemiptera by the development of a novel form of JH III containing a third epoxide link. This is a crucial event given the importance of JH in controlling many aspects of the development, reproduction and physiology of insects. While the paper achieves that objective, there are still some improvements that could be made.

I find the new description of the most recent view of the lineage of modern Heteroptera beginning at line 158 to be confusing. It is important to remember that the principal objective of this paper is to demonstrate that the production of JHSB3 occurred when the Heteroptera diverged from the other Hemiptera. For the purposes of this study, when in evolutionary time groups made their first appearance is not particularly relevant. The important and relevant issue is the relatedness of the major groups, the infraorders. Attempting to put this into a verbal description is not easy. Would it not be simpler to use Figure 2 as the primary reference and refer to shared ancestry? Thus, the two infra orders Dipsocoramorpha and Enicocephalomorpha that have not been examined for JHSB3 production share a common ancestry with Gerromorpha and Nepomorpha which produce JHSB3. While it is possible that Dipsocoramorpha and Enicocephalomorpha might have lost the capacity to produce the new JH, it is extremely unlikely, particularly since they do not share a common ancestor with each other. Given that all of the other lineages of Heteroptera produce the same JH, the assumption that the Dipsocoramorpha and Enicocephalomorpha also produce it is reasonable.

Line 189 indicates that the appearance of a new form of JH will require a simultaneous modification of the receptor. The evolutionary relationship between receptors and ligands is not as simple as might be implied by such a statement. Many compounds totally unrelated to JH will mimic the effects of JH, and it would be possible for a modified JH to bind to a receptor for unmodified JH III. These issues can only be resolved by properly conducted binding studies, well beyond the scope of the current study. While I can agree that some minor modification of an existing receptor might eventually arise, it is perhaps not an absolute prerequisite. Similarly the binding site(s) on any hemolymph JH binding protein (which protects JH from degradation before it reaches target tissues) may not require modification. Indeed, I have been unable to find any evidence in the literature for the existence of specific JH binding proteins in hemiptera. In some insects, the protective function resides in a general non-specific lipophorin. It would be wise to moderate the language of these two issues making it less of a requirement. If the authors have evidence that demonstrates the existence of a specific hemolymph JH binding protein in a heteropteran, they should include it. I confess to skepticism that such a protein has been demonstrated. If it has not, then that issue can be eliminated from the discussion or at the very least, modified.

Finally, it is important to recognize that this study deals, appropriately, with the product of the corpus allatum. Like almost every other publication in the field, it is assumed that the product of the ca is the effective hormone at the target sites. I have argued that this may not always be the case (*Insect Biochemistry and Molecular Biology* 30 (2000) 663–669) and that target tissues may modify the product of the ca. I don't think this requires changes in the current MS.

Review form: Reviewer 2**Is the manuscript scientifically sound in its present form?**

Yes

Are the interpretations and conclusions justified by the results?

Yes

Is the language acceptable?

Yes

Do you have any ethical concerns with this paper?

No

Have you any concerns about statistical analyses in this paper?

Yes

Recommendation?

Accept with minor revision (please list in comments)

Comments to the Author(s)

Dear authors, thank you for the revised manuscript. I have recommended it for publication with minor revision. Please see the attached file (Appendix A) for detailed comments. Best wishes for 2021!

Decision letter (RSOS-202242.R0)

Dear Dr Goto

On behalf of the Editors, we are pleased to inform you that your Manuscript RSOS-202242 "Juvenile hormone III skipped bisepoxide is wide-spread in true bugs (Hemiptera: Heteroptera)" has been accepted for publication in Royal Society Open Science subject to minor revision in accordance with the referees' reports. Please find the referees' comments along with any feedback from the Editors below my signature.

Please submit your revised manuscript and required files (see below) no later than 7 days from today's (ie 07-Jan-2021) date. Note: the ScholarOne system will 'lock' if submission of the revision is attempted 7 or more days after the deadline. If you do not think you will be able to meet this deadline please contact the editorial office immediately.

on behalf of Prof Kevin Padian (Subject Editor)
 openscience@royalsociety.org

Associate Editor Comments to Author:

The reviewers have a few further recommendations following the transfer of your paper to RSOS from PRSB - as these seem to be relatively straightforward (or at least requiring relatively limited time), we will be prepared to accept a final version of your paper if you are able to make the changes recommended.

Reviewer comments to Author:

Reviewer: 1

Comments to the Author(s)

The principal objective of this study is to demonstrate that the Heteroptera are distinguished from the remaining Hemiptera by the development of a novel form of JH III containing a third epoxide link. This is a crucial event given the importance of JH in controlling many aspects of the development, reproduction and physiology of insects. While the paper achieves that objective, there are still some improvements that could be made.

I find the new description of the most recent view of the lineage of modern Heteroptera beginning at line 158 to be confusing. It is important to remember that the principal objective of this paper is to demonstrate that the production of JHSB3 occurred when the Heteroptera diverged from the other Hemiptera. For the purposes of this study, when in evolutionary time groups made their first appearance is not particularly relevant. The important and relevant issue is the relatedness of the major groups, the infraorders. Attempting to put this into a verbal description is not easy. Would it not be simpler to use Figure 2 as the primary reference and refer to shared ancestry? Thus, the two infra orders Dipsocoramorpha and Enicocephalomorpha that have not been examined for JHSB3 production share a common ancestry with Gerromorpha and Nepomorpha which produce JHSB3. While it is possible that Dipsocoramorpha and Enicocephalomorpha might have lost the capacity to produce the new JH, it is extremely unlikely, particularly since they do not share a common ancestor with each other. Given that all of the other lineages of Heteroptera produce the same JH, the assumption that the Dipsocoramorpha and Enicocephalomorpha also produce it is reasonable.

Line 189 indicates that the appearance of a new form of JH will require a simultaneous modification of the receptor. The evolutionary relationship between receptors and ligands is not as simple as might be implied by such a statement. Many compounds totally unrelated to JH will mimic the effects of JH, and it would be possible for a modified JH to bind to a receptor for unmodified JH III. These issues can only be resolved by properly conducted binding studies, well beyond the scope of the current study. While I can agree that some minor modification of an existing receptor might eventually arise, it is perhaps not an absolute prerequisite. Similarly the binding site(s) on any hemolymph JH binding protein (which protects JH from degradation before it reaches target tissues) may not require modification. Indeed, I have been unable to find any evidence in the literature for the existence of specific JH binding proteins in hemiptera. In some insects, the protective function resides in a general non-specific lipophorin. It would be wise

to moderate the language of these two issues making it less of a requirement. If the authors have evidence that demonstrates the existence of a specific hemolymph JH binding protein in a heteropteran, they should include it. I confess to skepticism that such a protein has been demonstrated. If it has not, then that issue can be eliminated from the discussion or at the very least, modified.

Finally, it is important to recognize that this study deals, appropriately, with the product of the corpus allatum. Like almost every other publication in the field, it is assumed that the product of the ca is the effective hormone at the target sites. I have argued that this may not always be the case (*Insect Biochemistry and Molecular Biology* 30 (2000) 663–669) and that target tissues may modify the product of the ca. I don't think this requires changes in the current MS.

Reviewer: 2

Comments to the Author(s)

Dear authors, thank you for the revised manuscript. I have recommended it for publication with minor revision. Please see the attached file for detailed comments. Best wishes for 2021!

===PREPARING YOUR MANUSCRIPT===

===PREPARING YOUR REVISION IN SCHOLARONE===

To revise your manuscript, log into <https://mc.manuscriptcentral.com/rsos> and enter your Author Centre - this may be accessed by clicking on "Author" in the dark toolbar at the top of the

page (just below the journal name). You will find your manuscript listed under "Manuscripts with Decisions". Under "Actions", click on "Create a Revision".

<https://royalsociety.org/journals/authors/author-guidelines/#supplementary-material> to include a suitable title and informative caption. An example of appropriate titling and captioning may be found at https://figshare.com/articles/Table_S2_from_Is_there_a_trade-off_between_peak_performance_and_performance_breadth_across_temperatures_for_aerobic_sc_ope_in_teleost_fishes_/3843624.

Author's Response to Decision Letter for (RSOS-202242.R0)

See Appendix B.

Decision letter (RSOS-202242.R1)

Dear Dr Goto,

It is a pleasure to accept your manuscript entitled "Juvenile hormone III skipped bisepoxide is wide-spread in true bugs (Hemiptera: Heteroptera)" in its current form for publication in Royal Society Open Science.

on behalf of Professor Kevin Padian (Subject Editor)
openscience@royalsociety.org

Follow Royal Society Publishing on Twitter: @RSocPublishing
Follow Royal Society Publishing on Facebook:
<https://www.facebook.com/RoyalSocietyPublishing.FanPage/>

Read Royal Society Publishing's blog:
<https://royalsociety.org/blog/blogsearchpage/?category=Publishing>

Appendix A

Matsumoto et al, manuscript RSOS-202242

Juvenile hormone (JH) governs many aspects of insect development and reproduction. The native JHs consist of a group of closely related biomolecules utilizing the same sesquiterpenoid scaffold. The authors focus their work on so called JHIII skipped bisepoxide (JHSB3), a biologically active JH homolog discovered recently in several heteropteran species. They ask whether JHSB3 could be the innate hormone for the entire order of Heteroptera. To this point, they collect 31 “true bug” species, covering most of the heteropteran lineages, and analyze their JHSB3 production by an established UPLC-MS/MS method. They find that all the species tested produce JHSB3 and conclude that “the common ancestor of Heteroptera switched their innate JH from JHIII to JHSB3”.

General comments:

The work presented in this manuscript is a logical and meritorious extension of the previous research conducted in author’s laboratory. Although the result section is brief, it amounts to a considerable amount of field work for which the authors certainly deserve credit.

The analytical measurements are summarized in Figure 3 of the manuscript. It presents a series of UPLC-MS/MS analyses of the CA extracts of various heteropteran specimens with JHSB3 convincingly evident in all the samples. Although many of the measurements were performed on a very limited number of insect specimens, the authors claim in their response to Referee 1 that there was not a single case of internal discrepancy between them.

It seems unfortunate that the authors have deliberately narrowed the CA extract analysis to JHSB3, missing the opportunity to detect other potential JH forms (mainly the “ancestral” JHIII) in the sample. Their previous work showed that the well-established UPLC-MS/MS method is perfectly capable of detecting all the main JH forms in a single run (Ando et al, 2020). Indeed, the very same reference points out a simultaneous presence of JHIII and JHSB3 in the bean bug *Riptortus pedestris* serving as a good precedent for the above argument. It also demonstrates significant differences in JH content between male and female of the same species – the issue that was not addressed in the current manuscript, most likely due to a limited number of collected bug specimens.

The authors claim that JHSB3 is the innate hormone of Heteroptera only. This claim seems a bit speculative as they do not include representatives of other hemipteran branches into their analysis due to the restricted scope of the work. This might be an interesting topic for their future research.

Specific comment:

The last two structures drawn in Figure 1 have no relevance to the work in the manuscript. It would be better to remove them for clarity.

Conclusion:

Despite the limitations discussed above, **I do recommend** publication of this manuscript in the Royal Society Open Science.

Appendix B

Responses to Reviewers

RSOS Associate Editor Comments to Author:

The reviewers have a few further recommendations following the transfer of your paper to RSOS from PRSB - as these seem to be relatively straightforward (or at least requiring relatively limited time), we will be prepared to accept a final version of your paper if you are able to make the changes recommended.

- (RESPONSE) Thank you very much for recognizing the significance of our study. In the decision letter, we found the comments from two RSOS reviewers and the comments from the RSPB reviewer 1 that we had received from RSPB. You can find our responses to them in blue.

RSOS reviewer comments to Author:

Reviewer: 1

Comments to the Author(s)

The principal objective of this study is to demonstrate that the Heteroptera are distinguished from the remaining Hemiptera by the development of a novel form of JH III containing a third epoxide link. This is a crucial event given the importance of JH in controlling many aspects of the development, reproduction and physiology of insects. While the paper achieves that objective, there are still some improvements that could be made.

I find the new description of the most recent view of the lineage of modern Heteroptera beginning at line 158 to be confusing. It is important to remember that the principal objective of this paper is to demonstrate that the production of JHSB₃ occurred when the Heteroptera diverged from the other Hemiptera. For the purposes of this study, when in evolutionary time groups made their first appearance is not particularly relevant. The important and relevant issue is the relatedness of the major groups, the infraorders. Attempting to put this into a verbal description is not easy. Would it not be simpler to use Figure 2 as the primary reference and refer to shared ancestry? Thus, the two infra orders Dipsocoramorpha and Enicocephalomorpha that have not been examined for JHSB₃ production share a common ancestry with Gerromorpha and Nepomorpha which produce JHSB₃. While it is possible that Dipsocoramorpha and Enicocephalomorpha might have lost the capacity to produce the new JH, it is extremely unlikely, particularly since they do not share a common ancestor with each other. Given that all of the other lineages of Heteroptera produce the same JH, the assumption that the Dipsocoramorpha and Enicocephalomorpha also produce it is reasonable.

- (RESPONSE) Thank you very much for your idea to improve our manuscript. According to this

comment, we simplified our explanation on the evolutionary history of Heteroptera in the first paragraph of the discussion: “In the present study, we clarified that all species belonging to Pentatomomorpha, Cimicomorpha, Leptopodomorpha, Nepomorpha, and Gerromorpha produces JHSB₃ (Fig. 2), suggesting that the common ancestor of Heteroptera possessed JHSB₃ as an innate JH.”

Line 189 indicates that the appearance of a new form of JH will require a simultaneous modification of the receptor. The evolutionary relationship between receptors and ligands is not as simple as might be implied by such a statement. Many compounds totally unrelated to JH will mimic the effects of JH, and it would be possible for a modified JH to bind to a receptor for unmodified JH III. These issues can only be resolved by properly conducted binding studies, well beyond the scope of the current study. While I can agree that some minor modification of an existing receptor might eventually arise, it is perhaps not an absolute prerequisite. Similarly the binding site(s) on any hemolymph JH binding protein (which protects JH from degradation before it reaches target tissues) may not require modification. Indeed, I have been unable to find any evidence in the literature for the existence of specific JH binding proteins in hemiptera. In some insects, the protective function resides in a general non-specific lipophorin. It would be wise to moderate the language of these two issues making it less of a requirement. If the authors have evidence that demonstrates the existence of a specific hemolymph JH binding protein in a heteropteran, they should include it. I confess to skepticism that such a protein has been demonstrated. If it has not, then that issue can be eliminated from the discussion or at the very least, modified.

➤ (RESPONSE) We understand the reviewer’s point. According to this and the RSBP reviewer’s comments, we eliminated the issue from the discussion.

Finally, it is important to recognize that this study deals, appropriately, with the product of the corpus allatum. Like almost every other publication in the field, it is assumed that the product of the ca is the effective hormone at the target sites. I have argued that this may not always be the case (Insect Biochemistry and Molecular Biology 30 (2000) 663–669) and that target tissues may modify the product of the ca. I don’t think this requires changes in the current MS.

➤ (RESPONSE) This is an important point, and we must bear in mind. However, according to the last sentence of this comment, we did not change the current MS.

Reviewer: 2

Comments to the Author(s)

Dear authors, thank you for the revised manuscript. I have recommended it for publication with

minor revision. Please see the attached file for detailed comments. Best wishes for 2021!

➤ (RESPONSE) Thank you very much for encouraging us. We really appreciate.

Juvenile hormone (JH) governs many aspects of insect development and reproduction. The native JHs consist of a group of closely related biomolecules utilizing the same sesquiterpenoid scaffold. The authors focus their work on so called JHIII skipped bisepoxide (JHSB₃), a biologically active JH homolog discovered recently in several heteropteran species. They ask whether JHSB₃ could be the innate hormone for the entire order of Heteroptera. To this point, they collect 31 “true bug” species, covering most of the heteropteran lineages, and analyze their JHSB₃ production by an established UPLC-MS/MS method. They find that all the species tested produce JHSB₃ and conclude that “the common ancestor of Heteroptera switched their innate JH from JHIII to JHSB₃”.

General comments:

The work presented in this manuscript is a logical and meritorious extension of the previous research conducted in author’s laboratory. Although the result section is brief, it amounts to a considerable amount of field work for which the authors certainly deserve credit.

The analytical measurements are summarized in Figure 3 of the manuscript. It presents a series of UPLC-MS/MS analyses of the CA extracts of various heteropteran specimens with JHSB₃ convincingly evident in all the samples. Although many of the measurements were performed on a very limited number of insect specimens, the authors claim in their response to Referee 1 that there was not a single case of internal discrepancy between them.

It seems unfortunate that the authors have deliberately narrowed the CA extract analysis to JHSB₃, missing the opportunity to detect other potential JH forms (mainly the “ancestral” JHIII) in the sample. Their previous work showed that the well-established UPLC-MS/MS method is perfectly capable of detecting all the main JH forms in a single run (Ando et al, 2020). Indeed, the very same reference points out a simultaneous presence of JHIII and JHSB₃ in the bean bug *Riptortus pedestris* serving as a good precedent for the above argument. It also demonstrates significant differences in JH content between male and female of the same species – the issue that was not addressed in the current manuscript, most likely due to a limited number of collected bug specimens.

➤ (RESPONSE) Due to a limited number of collected specimens, we primarily focused on the presence/absence of JHSB₃ in the present study. However, as commented by this reviewer, it is important to investigate the presence/absence of JH III and the ratio of JHSB₃ and JH III in these heteropteran insects, although several papers have detected no JH III in several heteropteran species. We discussed this point in the third paragraph of the discussion in the

current MS. It is also important to compare the JHSB₃ hemolymph titre along with sexual maturation. However, the issue is far beyond the scope of this study, and thus, we did not mention this point in the current MS.

The authors claim that JHSB₃ is the innate hormone of Heteroptera only. This claim seems a bit speculative as they do not include representatives of other hemipteran branches into their analysis due to the restricted scope of the work. This might be an interesting topic for their future research.

- (RESPONSE) According to this and the RSBP reviewer's comments, we deleted the speculative statement and described our conclusion as "JHSB₃ is wide-spread in heteropteran bugs and the evolutionary occurrence of JHSB₃ ascends to the common ancestor of Heteroptera" in the abstract and "Thus, the evolutionary occurrence of JHSB₃ ascends to the common ancestor of Heteroptera" in the second paragraph of the discussion.

Specific comment:

The last two structures drawn in Figure 1 have no relevance to the work in the manuscript. It would be better to remove them for clarity.

- (RESPONSE) We appreciate this comment. We deleted the last two structures in Fig. 1.

Conclusion:

Despite the limitations discussed above, I do recommend publication of this manuscript in the Royal Society Open Science.

- (RESPONSE) Thank you very much for encouraging us. We appreciate.

RSPB reviewer

Comments to the Author(s)

- (RESPONSE) We received the following comments from the RSPB reviewer in the RSPB decision letter. We have responded to these comments and revised our manuscript according to them when we submitted our manuscript to RSOS. Here, we, once again, respond to these comments because the RSOS editor sent us the comment file again. However, we did not make track changes based on these comments, because we had already revised our manuscript when we submit the Ro version to RSOS.

Matsumoto et al, manuscript RSPB-2020-1299

Juvenile hormone (JH) governs many aspects of insect development and reproduction. The

native JHs consist of a group of closely related biomolecules utilizing the same sesquiterpenoid scaffold. The authors focus their work on so called JHIII skipped bisepoxide (JHSB₃), a biologically active JH homolog discovered recently in several heteropteran species. They are asking, whether JHSB₃ could be the innate hormone for the entire order of Heteroptera. To this point, they collect 32 “true bug” species, covering the majority of heteropteran lineages, and analyze their JHSB₃ production by an established UPLC-MS/MS method. They find that all the species tested produce JHSB₃ and conclude that “the common ancestor of Heteroptera switched their innate JH from JHIII to JHSB₃”.

The work presented in this manuscript builds on the previous research conducted in author’s laboratory. Although it seems like a logical and meritorious extension of their previous findings, the practical execution suffers from superficial evidence and limited scope:

The results section is covered in two brief paragraphs, corresponding to two main figures. The first figure is a phylogenetic tree adapted from other (referenced) source.

The second figure presents a series of raw chromatograms that should serve as a single and final evidence of JHSB₃ presence in the samples. Many data were collected from a single (!) insect specimen, there is no mention of statistics, quantification (as in ref. 39: Villalobos-Sambucaro et al., 2020) or reproducibility. Some LC traces are very noisy, most likely close to a detection limit of the given method. No MS analysis is presented to support the analytical work. No discussion of possible male/female differences, seen in their previous work (ref. 16: Ando et al., 2020) is provided.

➤ (RESPONSE) We collected insects from the field and immediately dissected out the CAs to obtain the CA products. Some insects were difficult to collect and the number of collected individuals was limited. Besides, some are very small and dissecting out the CA is very difficult. Due to the difficulty, some of our data are based on a single specimen. This was described in the first paragraph of the M & M in the manuscript. If we detected JHSB₃ in some specimens but not in other specimens, it is problematic because there is a possibility that the variance was caused by the small number of individuals used. However, in the present study, we detected JHSB₃ in all species with our highly sensitive UPLC-MS/MS analytical method. It is also important to note that the present study concentrated on the presence/absence of JHSB₃ because our primary focus is the evolutionary occurrence of this hormone, as mentioned in the third and fourth paragraphs of the Introduction. We did not quantify JHSB₃ because we did not regulate the reproductive status of the individuals used. It is well-known that the JH titre is closely related to the reproductive status of insects. We used field-collected adult bugs and immediately used for experiments, and thus we do not know their reproductive status and history. Therefore, even if the JH amount was high or low, we are not able to withdraw any information from the data. In the present study, we showed the

intensity of the JHSB₃ signal, instead of the amount of JHSB₃, which are comparable among samples. As pointed out by this reviewer, some LC traces were rather noisy because small CAs produced a small amount of JH. Even though, we detected a clear JHSB₃ peak in all LC traces. As Matsumoto et al. (2020) revealed, our chiral UPLC-MS/MS can detect 0.25 pg of JHSB₃. It is well-known that both males and females produce JH, but due to a limited number of specimens collected, we did not pay attention to the sexual difference.

The chromatograms seem cut-off at the fourth minute of analytical time, possibly precluding the evidence of other JH species (JHIII, JHI) present (compare to ref. 16: Ando et al, 2020). The presence/absence of JHIII and JHI in the samples is not discussed at all. Also, a non-expert reader would certainly appreciate a figure showing the chemical structures of JH homologues discussed in the manuscript.

- (RESPONSE) We realize that our figure legend made the reviewer being confused. We showed the signal intensity of the product ion at m/z 233.2 produced from the precursor ion at m/z at 283.2 in Fig. 2. Thus, even if we extend the analytical time further, we never detect JH III and JH I. Thus, it is obvious that we did not cut-off at the fourth min of analytical time to preclude the presence of JH III and JH I. We had modified the figure legend for the UPLC-MS/MS analysis. Because we concentrated on the presence/absence of JHSB₃, we did not analyze the presence of JH III and JH I in the present study. In the revised manuscript, we added a figure showing the chemical structures of JH homologues described in the manuscript.

The supplementary figure S1 showing photographs of the species used in the study is poorly formatted, the photographs are not referenced (as some are clearly taken from elsewhere), the size bars are not annotated thus losing its meaning.

- (RESPONSE) It is important to note that we took all the photographs. We used some ecological photographs to explain how the species look like. However, this did not allow us to add the scale bar for them. If you are unsatisfied with this, we are happy to remove the ecological photographs of *Aquarius paludum*, *Agriosphodrus dohrni*, *Riptortus pedestris*, and *Erthesina fullo* from the figure.

The authors failed to perform any kind of biological activity test to support their hypothesis (such as in ref.16: Ando et al, 2020).

- (RESPONSE) Biological activity of JHSB₃ has been examined in two heteropteran species, *Plautia stali*, and *Riptortus pedestris* (Ando et al., 2020; Kotaki et al., 2011; Lee et al., 2019). Because our primary focus is the presence/absence of JHSB₃, we did not investigate its

biological activity in the present study. Besides, it is almost impossible to investigate the biological activity in all the species used in the present study because most of them are collected from the field and the number of individuals collected from the field was very limited.

The authors claim that JHSB₃ is the innate hormone of Heteroptera only. However, they do not include representatives of any other hemipteran branches into their analysis. This makes their claim less solid and more speculative.

- (RESPONSE) It is important to note that the JH of non-heteropteran Hemiptera is established to be JH III in the previous studies (Bertuso and Tojo, 2002; Gelman et al., 2007; Hardie et al., 1985; Ishikawa et al., 2012; Schwartzberg et al., 2008). However, according to this and the RSOS reviewer comments, we deleted the speculative explanation and describe our conclusion as “JHSB₃ is wide-spread in heteropteran bugs and the evolutionary occurrence of JHSB₃ ascends to the common ancestor of Heteroptera” in the abstract and “Thus, the evolutionary occurrence of JHSB₃ ascends to the common ancestor of Heteroptera” in the second paragraph of the discussion. Significance of much wider taxon sampling in future study is now described in the third paragraph of the discussion in the current MS.

The authors argue that developing a new hormone analog also requires an adaptation of a JH-signaling system, including the structure of JH receptor. However, current literature provides evidence of JH receptors having similar affinity to several JH homologs. To better support their claim, the authors could attempt to assemble a sequence alignment of JH-binding PASB domain of JH-receptors of diverse species to see any conservative changes specific to heteropterans. It could be an interesting starting point of further research.

- (RESPONSE) Thank you very much for encouraging us. As you will find, the RSOS reviewer was seriously concerned with this point. Thus, we eliminated the part from the discussion in the current MS.